# Identification of the Maize LEA Gene Family and Its Relationship with Kernel Dehydration

**DOI:** 10.3390/plants12213674

**Published:** 2023-10-25

**Authors:** Yaping Zhang, Xiaojun Zhang, Liangjia Zhu, Lexin Wang, Hao Zhang, Xinghua Zhang, Shutu Xu, Jiquan Xue

**Affiliations:** Key Laboratory of Biology and Genetic Improvement of Maize in Arid Area of Northwest Region, College of Agronomy, Northwest A&F University, Yangling 712100, China; 2021050118@nwafu.edu.cn (Y.Z.); 13021695171@163.com (X.Z.); 18330850923@163.com (L.Z.); 13022965186@163.com (L.W.); zzhanghao@nwafu.edu.cn (H.Z.); zhxh4569@163.com (X.Z.)

**Keywords:** maize, ZmLEA, kernel dehydration rate, expression pattern

## Abstract

Maize, the most widely planted and highest yielding of the three major crops in the world, requires the development and breeding of new varieties to accommodate the shift towards mechanized harvesting. However, the moisture content of kernels during harvest poses a significant challenge to mechanized harvesting, leading to seed breakage and increased storage costs. Previous studies highlighted the importance of LEA (Late Embryogenesis Abundant) members in regulating kernel dehydration. In this study, we aimed to gain a better understanding of the relationship between the LEA family and grain dehydration in maize. Through expression pattern analysis of maize, we identified 52 *LEA* genes (*ZmLEA*s) distributed across 10 chromosomes, organized into seven subgroups based on phylogenetic analysis, gene structure, and conserved motifs. Evolutionary and selective pressure analysis revealed that the amplification of *ZmLEA* genes primarily resulted from whole-genome or fragment replication events, with strong purifying selection effects during evolution. Furthermore, the transcriptome data of kernels of two maize inbred lines with varying dehydration rates at different developmental stages showed that 14 *ZmLEA* genes were expressed differentially in the two inbreds. This suggested that the *ZmLEA* genes might participate in regulating the kernel dehydration rate (KDR) in maize. Overall, this study enhances our understanding of the ZmLEA family and provides a foundation for further research into its role in regulating genes associated with grain dehydration in maize.

## 1. Introduction

Maize is the most widely planted and highest yielding cereal crop in the world, including in China [1]. As production shifts from manual to mechanized operations due to changes in production methods and labor shortages, there is a need for maize varieties suitable for mechanized management, including sowing, watering, fertilization, and harvesting [2]. According to the State Ministry of Agriculture and Rural Affairs (https://www.moa.gov.cn/ (accessed on 20 December 2020)), China’s comprehensive mechanized production rate of maize reached 89.76% in 2020, with mechanized sowing rates (ratio of mechanically sown area to total cultivated area) above 90%. However, the mechanized harvesting rate (ratio of mechanically harvested area to cultivated area) remains below 85%, with most of the maize harvested as ears rather than kernels, resulting in increased costs for air-drying and storage [3]. Therefore, the development of new maize varieties suitable for mechanized kernel harvesting is essential. Currently, most maize varieties have 25–40% of kernel moisture content (KMC), which exceeds the standard KMC (<25%) required for kernel mechanized harvesting [4]. KMC is considered a limiting factor for mechanized harvesting [5]. Thus, understanding the genetic basis of KMC is crucial for breeding kernel varieties suitable for mechanized harvest in the genomic era.

During maize kernel development, the KMC at harvest is determined by the kernel dehydration rate (KDR) before and after physiological maturity [6]. According to a previous study by our group, one *LEA* (Late Embryogenesis Abundant) gene (*Zm00001eb238300*) was identified as being associated with the KDR by multiple omics [7]. Additionally, in the model plant Arabidopsis, LEA proteins have been reported to regulate kernel development and dehydration processes, especially members of the LEA_4 subfamily [8]. The LEA proteins were first isolated from cotyledons during embryonic development and were found to accumulate during seed maturation in cotton in 1981 [9], which was considered an indicator of seed maturity [10]. Similar proteins have been found in other plants, including Arabidopsis [8], sorghum [11], rice [12], and wheat [13]. They are rapidly synthesized and accumulate when plants encounter adverse environmental conditions, such as low temperatures, drought, and dehydration [14]. Normally, LEA proteins exist in an unstructured conformation under hydration conditions but adopt a structured conformation when an organism experiences water scarcity [15], allowing it to adapt to adverse environments.

Moreover, LEA proteins participate in various regulatory networks in Arabidopsis. Firstly, they interact with sugar materials to create a glassy state within cells, preventing excessive seed dehydration [16]. Secondly, as molecular chaperones, LEA proteins stabilize protein structures, protecting other proteins and membranes from aggregation, which is crucial for stress tolerance, especially dehydration and low-temperature stress [17]. Additionally, LEA proteins enhance stress tolerance by buffering the increase in cellular ions, regulating both internal and external cell osmotic potentials [18]. In summary, LEA proteins function as dehydration defenders, regulating water in plants during maturation, dehydration, and stress responses [19].

Based on multi-omics analyses, including genome-wide association analyses, transcriptomics, and proteomics, our team discovered that LEA family members are involved in regulating KMC and KDR during different development stages in maize kernels [7]. Previous studies also showed that abscisic acid (ABA) enhances the dehydration tolerance during maize kernel dehydration by regulating the abundance of LEA proteins [20]. Therefore, we conducted a systematic analysis of the LEA family in maize, including family identification and evolution, based on the B73 reference genome (http://maizegdb.org (accessed on 20 December 2022), Zm-B73-REFERENCE-NAM-5.0). Moreover, transcriptome data were collected to identify differentially expressed maize *LEA* genes (*ZmLEA*s) in two maize inbred lines with significantly different dehydration rates during late seed development. Further, the proteome data of kernels in five periods for the line KA105 from our lab were used to analyze the time-solved accumulation of ZmLEA proteins in maize. The aim of this study was to provide a theoretical basis for understanding the role of LEA family members in regulating KMC and KDR during maize kernel development and to guide germplasm improvement for mechanized maize kernel harvesting.

## 2. Materials and Methods

### 2.1. Identification of ZmLEA Genes

The maize genome data were downloaded from Maize GDB (http://maizegdb.org (accessed on 20 December 2022)), sorghum and rice *LEA* gene sequences from NCBI (https://ncbi.nlm.nih.gov (accessed on 20 December 2022)), and Arabidopsis *LEA* family gene sequences from the online database TAIR (https://arabidopsis.org (accessed on 20 December 2022)). To identify *ZmLEA*s, we extracted the published LEA protein sequences from Arabidopsis, rice, and sorghum as references and compared them with the B73 reference genome (version 5.0), using e-values (e < E^−5^) to obtain the maize LEA family members with TBtools software (version 1.112) [21]. Subsequently, to decrease the false positives, the retrieved candidate genes were compared again with entries in the SwissProt database (https://expasy.org/resources/uniprotkb-swiss-prot (accessed on 20 December 2022)). Finally, the conserved structural domains of the proteins were further screened using CD-Search and Pfam with e (e < E^−10^) as the threshold value, and 52 *ZmLEA*s were obtained after filtering out redundant genes.

### 2.2. Multiple Sequence Alignment and Phylogenetic Analysis

The acquired ZmLEA protein sequences of maize were aligned with multiple amino acid sequences from maize, Arabidopsis, rice, and sorghum using the ClustalW tool. The comparison findings were utilized to build ML (Maximum Likelihood) phylogenetic trees for the four species using MEGA11 (Bootstrap: 1000; alternative model: WAG). The evolutionary tree was visualized using the online tool EvolView (http://www.evolgenius.info/evolview.html (accessed on 22 December 2022)).

### 2.3. Characterization of the *ZmLEA* Proteins

The ExPASy website (https://web.expasy.org/protparam/ (accessed on 20 December 2022)) was used to analyze and physically characterize the molecular weight, isoelectric point, and overall average hydrophilicity of the ZmLEA proteins. Then, the BUSCA annotation system (https://busca.biocomp.unibo.it/ (accessed on 20 December 2022)) was used to predict the subcellular location of each ZmLEA protein.

### 2.4. Chromosome Localization and Homology Analysis

The chromosome locations of the *LEA* gene sequences in Arabidopsis, sorghum, and rice were obtained from TAIR and NCBI (https://www.ncbi.nlm.nih.gov (accessed on 20 December 2022)). The related LEA proteins were annotated in the Ensembl Plants databases. Interspecies covariance analysis and visualization were performed for *LEA* genes from maize, sorghum, rice, and Arabidopsis using the software programs TBtools (version 1.112) [21] and MCScanX (http://chibba.pgml.uga.edu/mcscan2/#tm (accessed on 20 December 2022)) [22].

### 2.5. Analysis of Gene Structure, Conserved Motifs, and cis-Acting Elements

The conserved motifs of ZmLEA proteins were examined using the online MEME software (https://meme-suite.org/meme/tools/meme (accessed on 20 December 2022)). The gene structures and protein conserved motifs of *ZmLEA* genes were visualized and detected using the TBtools software [21]. The cis-acting elements were predicted by subjecting the 2000 bp sequence upstream of the *ZmLEA* promoters to PlantCARE (https://bioinformatics.psb.ugent.be/webtools/plantcare/html/ (accessed on 20 December 2022)) and then visualized using the TBtools software (version 1.112).

### 2.6. GO Annotation Analysis

The amino acid sequences of the ZmLEA proteins were submitted to the online software eggNOG (http://eggnog-mapper.embl.de/ (accessed on 20 December 2022)), followed by GO annotation and enrichment analysis conducted using TBtools software for the ZmLEA genes. 

### 2.7. Transcriptome and Proteome Data Collection and Analysis

The transcriptome and proteome data were obtained from a previous study [7]. The fast-dehydrating maize inbred line (KA105) and the slow-dehydrating maize inbred line (KB020) were used as the study materials. Their average KMC at different kernel development times, determined over two years (2018 and 2019), is shown in Appendix A. The two inbred lines were planted in June 2018 at the Guancun Maize Experimental Base of Northwest Agriculture and Forestry University, Yangling District, Shaanxi Province, China (E 34°32′, N 108°05′). A completely randomized block experimental design was adopted, with a row length of 4.5 m, a row spacing of 0.6 m, four rows planted in each plot, two replications, and a density of 75,000 plants hm^−2^, and the field management conditions were the same as those used locally. All ears were bagged at the pre-sprouting stage and then hand-pollinated on the same date to minimize the environment noise. Kernels in the middle of well-pollinated ears at 14, 21, 28, 35, 42, 49, 56, and 63 days after pollination were sampled immediately for two duplicates—one was used to determine the KMC using an oven at 35 °C, and the other was frozen in liquid nitrogen. All samples were stored in a freezer at −80 °C for RNA and protein extraction. Three replicates were set up for sample at each time. Based on parameters including the false discovery rate (FDR), an adjusted *p*-value of 0.05, and a fold change (FC) of 2 or 0.5 (namely, −log_2_(FC) > 1) using edgeR, the differentially expressed genes were screened [23]. A differential gene expression heatmap and a differential protein expression heatmap were created using the OmicShare tool (https://www.omicshare.com/tools (accessed on 20 December 2022)).

## 3. Results

### 3.1. Identification of the Members of ZmLEA Family in Maize

To obtain candidate *ZmLEA* genes in maize, we compared maize protein sequences with known LEA protein sequences from three other species: Arabidopsis, sorghum, and rice. Then, the candidate *ZmLEA* genes were compared again with entries in the SwissProt database (https://expasy.org/resources/uniprotkb-swiss-prot (accessed on 20 December 2022)). After eliminating the redundant genes, 52 genes with conserved structural domains were retained, which were sequentially named *ZmLEA1–ZmLEA52* according to the distribution order of these genes on the 10 chromosomes of the B73 reference genome (Table 1).

Most of these 52 *ZmLEA* genes were harbored on chromosomes 1 and 3, which contained 9 *ZmLEA* genes each, while chromosome 9 contained only 1 *ZmLEA* gene. Notably, the *ZmLEA* genes showed a preferential distribution at both sides of the chromosomes, and some regions had two or four assembled *ZmLEA* genes. For example, *ZmLEA35–28* were located on the same region at the end of long arm on chromosome 6 (Figure 1). Further, the 52 *ZmLEA* genes were divided into seven subfamilies based on sequence homology and conserved motifs in the Pfam database, i.e., ZmLEA_1, ZmLEA_2, ZmLEA_3, ZmLEA_4, ZmLEA_5, dehydrins (DHN), and seed maturation protein (SMP) subfamilies [8]. 

### 3.2. Evolutionary Analysis of LEA Family Genes 

To analyze the evolution process of *LEA* family genes, phylogenetic trees were constructed using genes from maize, Arabidopsis, rice, and sorghum (Figure 2). The evolutionary results showed that LEA_2, with 30 members, was the most represented in maize, as well as in the other three species. Interesting, only one gene (*Sb02g028010*) from sorghum was found in the LEA_6 subfamily. A comparison of the *LEA* genes in the four species revealed greater similarity among monocotyledon species (maize, sorghum, and rice) than with the dicotyledon Arabidopsis (Figure 3A). Maize had 59 and 48 *LEA* homologous gene pairs compared to sorghum and rice, respectively, and only 13 *LEA* homologous gene pairs with Arabidopsis. Specifically, the sorghum genome contained 42 homologous *ZmLEA* genes, the rice genome 38 homologous *ZmLEA* genes, and the Arabidopsis genome only 8 homologous *ZmLEA* genes. This homology between maize and the other three species indicates that *LEAs* evolved before the division of monocotyledons and dicotyledons, and this evolution occurred concurrently with species evolution. Furthermore, the highest homology was observed between maize and sorghum, which is consistent with the evolution of these two species.

As maize is a paleo-tetraploid plant, gene duplications are normal. Here, we found that there were 18 collinear genes among the 52 *LEA* family genes in the maize genome, which were found to be distributed on chromosomes 1, 3, 5, and 8 (Figure 4). These collinear genes belong to the four subfamilies LEA_2, LEA_3, SMP, and DHN. Among them, the LEA_2 subfamily appeared to comprise the largest proportion, with 12 *LEA* genes (40% in LEA_2), while the three remaining subfamilies only had 2 collinear genes each. This indicates that gene duplication also contributed to the expansion of the maize LEA family. In addition, we calculated the Ka (the ratio of the number of synonymous substitutions per synonymous sites) and Ks (the ratio of the number of non-synonymous substitutions per non-synonymous sites) values of the 18 collinear gene pairs in maize, which indicated the direction of evolution. The Ka/Ks ratio of these collinear gene pairs was consistently <1, ranging from 0.21 (*ZmLEA13/51*) to 0.85 (*ZmLEA14/42*), with an average value of 0.50 (Figure 3B). Notably, *ZmLEA14/42* displayed a high Ka/Ks ratio (0.85), suggesting a rapid post-replication evolution for this gene pair. Overall, the Ka/Ks analysis indicated a strong purifying selection effect in the maize LEA family, aiding in the elimination of environmental disadvantages resulting from non-synonymous mutations. 

### 3.3. Physicochemical Characterization of Maize ZmLEA Genes

To better understand the ZmLEA family, the physicochemical characteristics were analyzed on the ExPASy website (Table 1). For the 52 *ZmLEA* gene sequences, the length ranged from 393 to 3269 base pairs (bp), except for the longest gene (*ZmLEA45*), which was 12,839 bp. As for the related encoded protein sequences, most contained fewer than 400 amino acids (aa), except for ZmLEA29, containing 604 aa, and ZmLEA47, containing 470 aa. Moreover, 65% (34) of the ZmLEA proteins had relatively high isoelectric points (PI > 7), especially in the LEA_2 subfamily. Regarding the hydrophilic mean values, the members of the subfamilies exhibited low hydrophilic mean values (GRAVY < 0), with the exception of half of the members of the LEA_2 subfamily. Furthermore, the DHN and SMP subfamilies showed strong hydrophilicity, with a low cysteine and a high lysine content in their amino acid sequences.

In addition, the subcellular location of different ZmLEA subfamilies was predict using the BUSCA annotation system (https://busca.biocomp.unibo.it/ (accessed on 20 December 2022)), and distinct subcellular locations were found. The proteins of the LEA_3 subfamily were found in mitochondria, while the LEA_5 proteins appeared localized in the nucleus. Other subfamilies showed localization in more than one cell region. For LEA_2, 43% of the proteins were nucleus-localized, with most having high PI (>7) and substantial hydrophobicity. This suggests that various ZmLEA subfamily proteins serve diverse functions in different organelles.

### 3.4. Prediction Analysis of the Structure and Function of ZmLEAs

Using the obtained protein sequences of the 52 *ZmLEA* genes, we extracted the gene structure of all genes according to the B73 reference genome version 5.0, based on the maizegdb database (http://www.maizegdb.org (accessed on 20 December 2022)). We found that members of the same subfamily had similar exon counts and exon and intron structures, but considerable variance existed between subfamilies (Figure 5). The majority of ZmLEA_2 subclade genes are around 2000 bp in length and have only one intron, except for ZmLEA9 (3269 bp), with two introns.Further, the conserved motifs in the ZmLEA proteins were analyzed, which also showed remarkably similar motifs within a given subfamily but diversities between different subfamilies (Figure 6). Due to fewer members and shared motifs occurring within the LEA_1 (*Zm00001eb308610*) and LEA_5 (*Zm00001eb286150* and *Zm00001eb286170*) subfamilies, the motifs in the genes of these two subfamilies are not presented in the results.

Meanwhile, the cis-acting elements of these 52 *ZmLEAs* were predicted using the 2000 bp sequence upstream of the promoter of each gene. Nine cis-acting elements were primarily identified, i.e., auxin responsiveness, light responsiveness, gibberellin responsiveness, ABA (abscisic acid) responsiveness, MeJA (methyl jasmonate) responsiveness, SA (salicylic acid) responsiveness, low-temperature responsiveness, defense and stress responsiveness, and wound responsiveness elements (Figure 7A). The light responsiveness elements occurred more than twice in all genes, but other elements were only randomly present in a few genes. In sum, two types of elements were noted: one responded to adverse environments, and the other responded to hormones, including ABA, MeJA, and SA. Lastly, a GO analysis was conducted to annotate the gene functions; these genes showed enrichment for defense responses (Figure 7B), which is consistent with the cis-element analysis. This implies that the *ZmLEA* gene family members react to stress and adversity.

### 3.5. Expression and Protein Abundance in Two Maize Inbred Lines with Different Kernel Dehydration Rates

To validate the relationship between *ZmLEA* expression patterns and protein abundance with KMC and KDR, we collected and analyzed the transcriptome and proteome data of kernels obtained in a previous study in our lab [7]. For the transcriptome data, the kernels of two inbreds, analyzed 21, 28, 35, 42, 49, and 56 days after pollination (DAP), were used to screen the differentially expressed *ZmLEA* genes. The first inbred line was KA105, which shows rapid dehydration, and the other inbred line was KB020, with the characteristic of slow dehydration (Appendix A). We found 14 *ZmLEA* genes with differential expression during kernel development, especially at DAP42, DAP49, and DAP56 (Figure 8). In addition, the expression levels of the *ZmLEA* genes changed with the kernel development, and most showed an upwards trend (Appendix A). For example, *ZmLEA9* and *ZmLEA19* exhibited an obviously declining trend in expression throughout kernel development in both inbred lines, while the other 12 *ZmLEA* genes accumulated with kernel development and displayed a significantly increasing trend. Comparing the expression of *ZmLEA* genes during the same period in the two maize inbred lines with different dehydration rates, the *ZmLEA27*, *ZmLEA34*, and *ZmLEA46* genes showed significantly higher expression in the slow-dehydrating inbred line (KB020) than in the fast-dehydrating inbred line (KA105). Conversely, the remaining 11 genes exhibited significantly higher expression in KA105 compared to KB020. These findings suggest that the *ZmLEA* gene family plays an essential role in seed development, particularly in the later stages.

For the proteome data, the kernels of KA105 at DAP28, DAP35, DAP42, DAP49, and DAP56 were collected, and 15 ZmLEAs showed differential abundance (Figure 9). Among them, 13 proteins, including ZmLEA1, tended first to decrease and then to increase, with DAP35 being a turning point. ZmLEA43, ZmLEA44, and ZmLEA48 showed a trend of decreased expression at DAP49. This is consistent with the trends observed in the transcriptome data. In addition, ZmLEA48 showed a consistently high expression, while ZmLEA20 showed a lower protein expression at DAP28 and DAP42. Based on this comprehensive transcriptome and proteome analysis, we suggest that ZmLEAs participate in regulating the KMC and KDR at the transcript or protein level. 

## 4. Discussion

The *LEA* gene family is vast and complex, present not only in common crops like wheat, sorghum, and rice, but also in vascular and non-vascular plants, fungi, bacteria, and invertebrates [24]. In a model plant, the *LEA* gene family has been understood more clearly than in other species [25]. To better understand the *LEA* gene family in maize, which is the most widely planted and highest yield crop in the world, we conducted a comparative genome analysis to identify *LEA* members in maize. In a previous study [26], Li and Cao identified 32 *ZmLEA* genes and divided them into nine subfamilies based on the B73 reference genome version 3.0. Considering the high accuracy and integrity of the B73 reference genome version 5.0, we identified 52 *ZmLEA* genes in maize, which were categorized into seven subfamilies by comparison of the B73 reference genome (version 5.0) with the identified *LEA* genes in Arabidopsis, rice, and sorghum. In Li and Cao’s study [26], some *ZmLEA*s were not annotated correctly, which resulted in missing some *ZmLEA*s, such as *ZmLEA*29 (*Zm00001eb238300*), which was identified in this study but not in theirs. Additionally, the sole member of the ZmLEA_6 subfamily (*GRMZM2G322351* or *Zm00001eb316370*) was not defined as an *LEA* family member. The results of this study indicated that the highest proportion of genes was found in the ZmLEA_2 subfamily, containing 30 members, while the lowest proportion was found in the ZmLEA_1 subfamily, which only contained 1 gene (Table 1). By analyzing the phylogenetic relationship of the *LEA* genes in these four species, we found the LEA_6 subfamily only existed in sorghum. It was also reported that the LEA_6 subfamily is not found in the algal and rice genomes [27,28]; it now can be stated as missing in maize. The results of the covariance analysis in four species—maize, sorghum, rice, and Arabidopsis thaliana—showed that the *LEA* family was mainly represented by the LEA_2 subfamily and the LEA_4 subfamily. 

According to the physicochemical characterization of the *ZmLEA* genes, we found most of the ZmLEA proteins were hydrophilic, according to their hydrophilicity mean (GRAVY) values (Table 1), similar to the LEA proteins in *Arabidopsis* and *Brassica napus* [8,29]. Many studies showed that the high hydrophilicity can be due to the disordered nature of the LEA proteins in their natural state, which is why the LEA proteins are also referred to as disordered proteins [30]. The 30 members of the ZmLEA_2 subfamily were found to include both hydrophilic and hydrophobic proteins, a result that is consistent with previous studies on Arabidopsis, rice, etc. [8,27]. The genes of the LEA_2 subfamily are considered to have low homology with those of other subfamilies, as shown by their gene structure, which could explain the atypical features of the LEA_2 subfamily members (Figure 5). Furthermore, most members of the *ZmLEA* family contain only one intron and have a relatively short length (Figure 5). This is consistent with previous studies showing that stress response-related genes have almost no introns [31,32]. It is also consistent with the gene structures of *LEA* genes that have been reported in other plants. For instance, ~60% of *the LEA* genes in wheat lack introns, while over half of the *LEA* genes in *Arabidopsis* contain only one intron per member [33]. From the point of view of the plant body stress response, fewer intronic structures reduce the time from transcription to translation in the defense against stress, lowering the cost of energy consumption and facilitating the plant’s rapid response to stimuli caused by adverse environmental conditions [34].

Through time-resolved research of LEA gene transcription and polypeptides in *Arabidopsis*, it was found that most LEA polypeptides appeared in the final stages of seed maturation, while their transcripts were detected at 10–20 days of maturation [35]. Here, we collected transcriptome and proteome data and found that *ZmLEA* gene expression was higher than ZmLEA peptide expression during the maize kernel development process, especially in the late stages of kernel maturation. This is consistent with the conclusion that the accumulation of LEA is limited when a plant acquires desiccation tolerance [35]. *SbLEA3A-1* and *SbLEA3A-2* in sorghum are highly homologous to *ZmLEA20*, and *SbLEA3B-2* and *ZmLEA2* are homologous genes belonging to the LEA_4 subfamily [36]. Interestingly, the *SbLEA3A* gene was expressed at a low level in developing seeds, with a significant increase in expression at maturity. *SbLEA3B-2* was expressed only at maturity. This is largely consistent with the transcriptome results of *ZmLEA2* expression during seed maturation. This unique expression pattern of the LEA_4 subfamily suggests that these genes may have specific roles at different stages of seed development. The transcriptome data of the kernels of two typical maize inbred lines at different developmental periods showed that 14 differentially expressed genes were identified among the 52 *ZmLEA* genes, of which 12 *ZmLEA* genes accumulated in large quantities in the late stage of maize kernel maturation, and two *ZmLEA* genes accumulated during kernel development. The expression tended first to increase and then to significantly decrease. In addition, the expression of the *ZmLEA27* and *ZmLEA34* genes showed an opposite trend compared to that of the other differentially expressed genes: in the slow-dehydrating type KB020 inbred line, it was significantly higher than in the fast-dehydrating KA105 inbred line. This difference in expression suggests that the expression of *ZmLEA* genes is highly related to the type of inbred line and might correspond to the rate of dehydration at the late stages of maize kernel development.

As for the function of the LEA proteins, it was demonstrated that they are involved in a variety of developmental processes and accumulate in large quantities when a plant faces drought, low temperature, salt stress, or is treated with the phytohormone ABA [37]. Under dehydration conditions, the LEA proteins can interact with phospholipids to help maintain the integrity of liposomes or with polypeptides as molecular chaperones that can inhibit the fusion of liposomes under desiccation stress and can protect proteins from dehydration-induced aggregation [38]. For example, proteins of the LEA_4 subfamily will form specific secondary structures to protect a variety of membrane structures and stabilize the cell under adverse environmental stresses, including dehydration stress and low-temperature stress [39,40,41]. In this study, the gene *Zm00001eb238300*, belonging to the ZmLEA_4 subfamily, was found to be located on chromosome 5 (chr5:145131834-145134200) and was defined as *ZmLEA29*, with gene ID *Zm00001d016128* in the B73 reference genome version 4.0. It was associated with KMC and KDR by genome-wide association [7]. Moreover, its varying expression at the transcriptome and proteome level also showed it would respond to KMC and KDR at different kernel development stages. With the exception of ZmLEA29, the other four members of the ZmLEA_4 subfamily, i.e., *ZmLEA2*, *ZmLEA12*, *ZmLEA20*, and *ZmLEA48*, were also found to be expressed differently at the transcription or protein level (Figure 8). This suggests the ZmLEA_4 subfamily responds to dehydration in maize to some extent. 

Given the inevitability of mechanized maize kernel harvesting development, we need to improve the plant density to achieve high yields, increase the kernel quality, and reduce the cost and labor requirements and breed new inbred lines with low KMC and fast KDR [42]. The application of molecular technology is becoming more and more effective in plant breeding; it can improve target traits directionally, such as herbicide resistance and insect resistance in maize [43,44]. However, few genes have been validated as affecting KMC or KDR by gene editing or mutants, besides *GAR2* (*GRMZM2G137211*), *CRY1-9* (*GRMZM5G805627*), and *ZmHSP5* (*Zm00001d047799*) [7,45], which limits their application in breeding. More molecular genetic and biochemic work needs to be performed to uncover the regulation mechanisms of KMC and KDR. It will be important to identify the dominant alleles in natural populations, explore functional markers to characterize phenotypes, clearly identify the up- or down-regulation of genes to construct regulation networks, and so on. This study confirmed the relationship between *ZmLEA* and KMC and KDR, suggesting some candidate genes (especially, genes from the *ZmLEA_4* subfamily) for optimizing the KMC and KDR in future maize breeding. 

## Figures and Tables

**Figure 1 plants-12-03674-f001:**
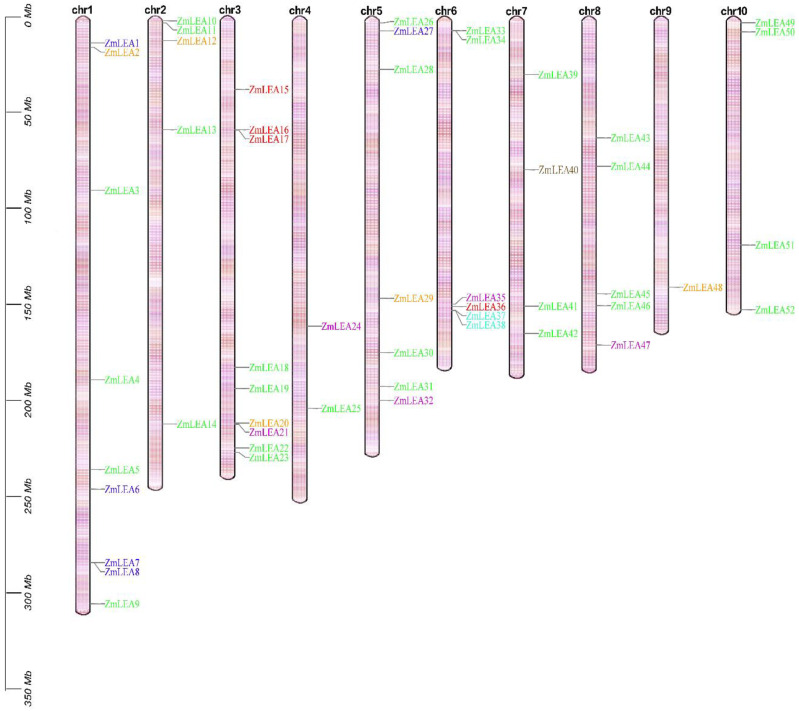
Distribution of the 52 *ZmLEA* genes on chromosomes in maize.

**Figure 2 plants-12-03674-f002:**
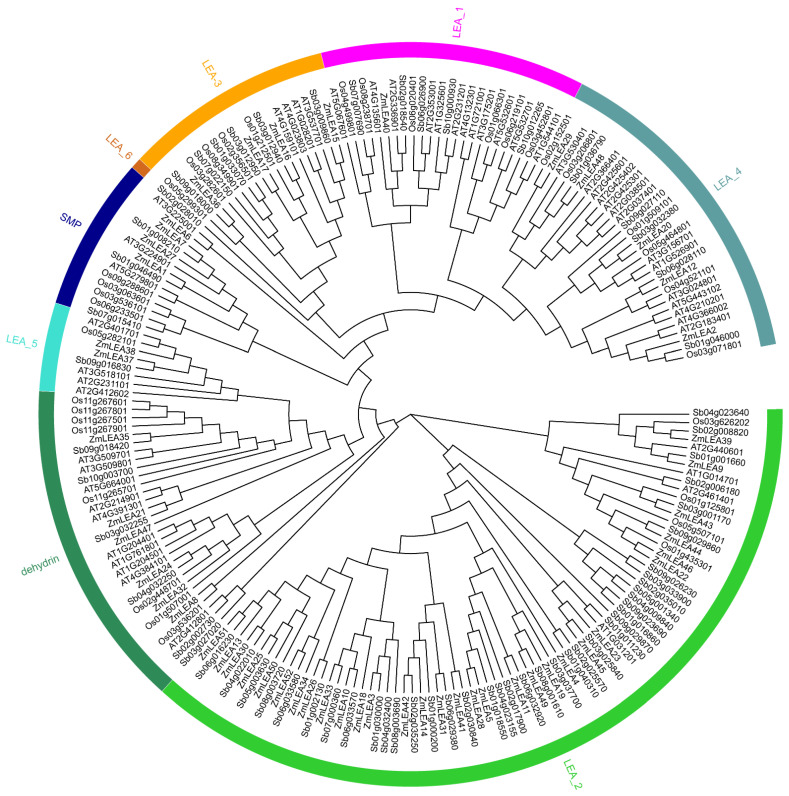
Evolutionary analysis of the *LEA* genes in maize, Arabidopsis, sorghum, and rice.

**Figure 3 plants-12-03674-f003:**
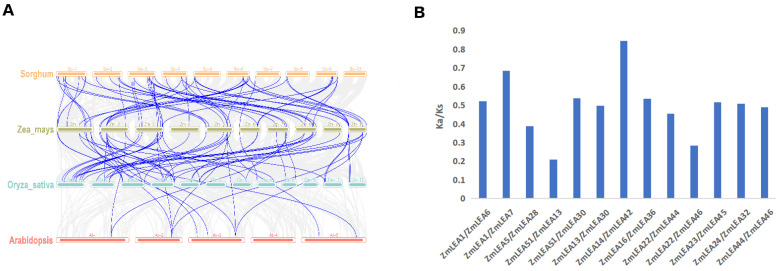
Co-lineage and gene selection analysis of *LEA* family genes. (**A**) Co-lineage analysis of *LEA* family genes in maize, sorghum, rice, and A. thaliana. The gray line in the Figure refers to the covariate gene pairs among the four species, and the blue line represents the covariate gene pairs with maize *LEA* genes in the genomes of each species. (**B**) Maize *LEA* family gene selection pressure analysis, *Ka* stands for synonymous substitutions, and *Ks* for non-synonymous substitutions.

**Figure 4 plants-12-03674-f004:**
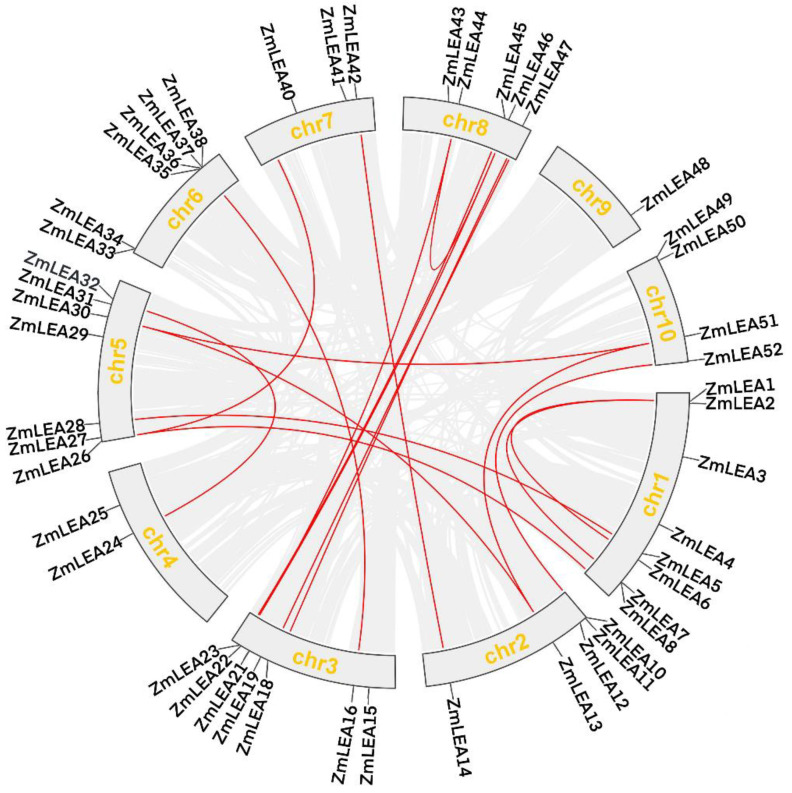
Covariate analysis of *ZmLEA* genes. The gray lines represent the covariates of the maize genome, and the red lines represent the *ZmLEA* covariate gene pairs.

**Figure 5 plants-12-03674-f005:**
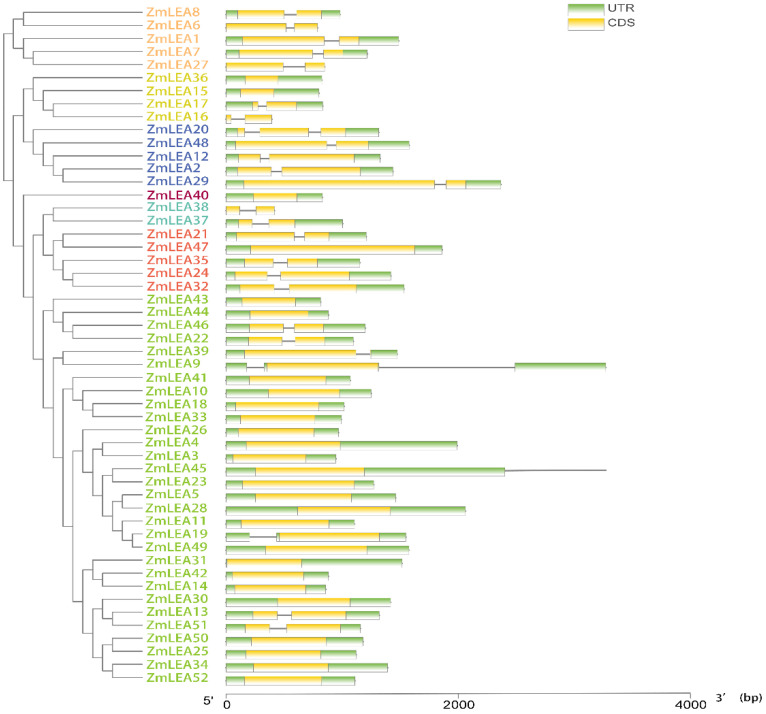
Gene structure of the *ZmLEA* genes. The colored boxes represent exons, and the gray lines represent introns.

**Figure 6 plants-12-03674-f006:**
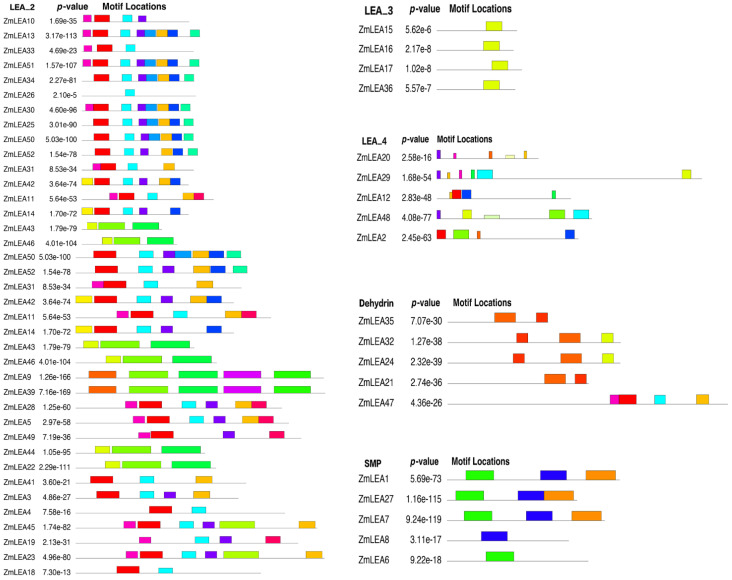
Conserved motif map of the *ZmLEA* genes.

**Figure 7 plants-12-03674-f007:**
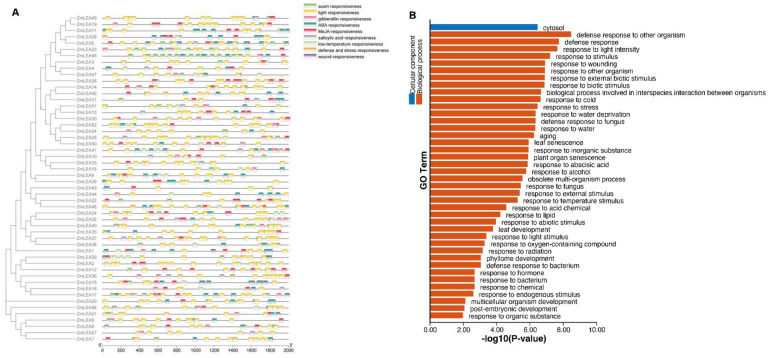
(**A**) Promoter cis-element analysis and (**B**) functional annotation of the *ZmLEA* genes.

**Figure 8 plants-12-03674-f008:**
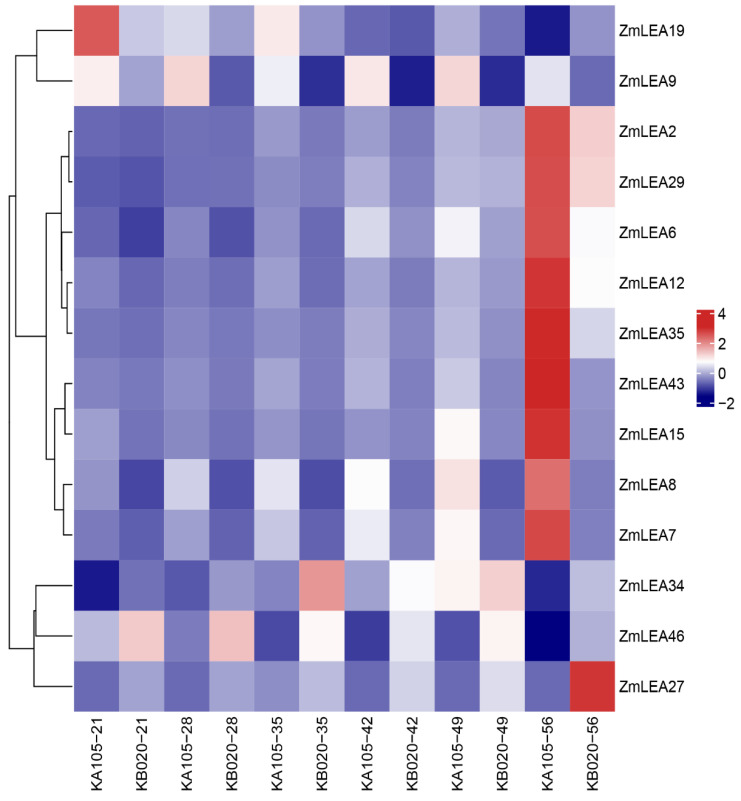
Expression heatmap of differentially expressed *ZmLEA* genes in KA105 and KB020.

**Figure 9 plants-12-03674-f009:**
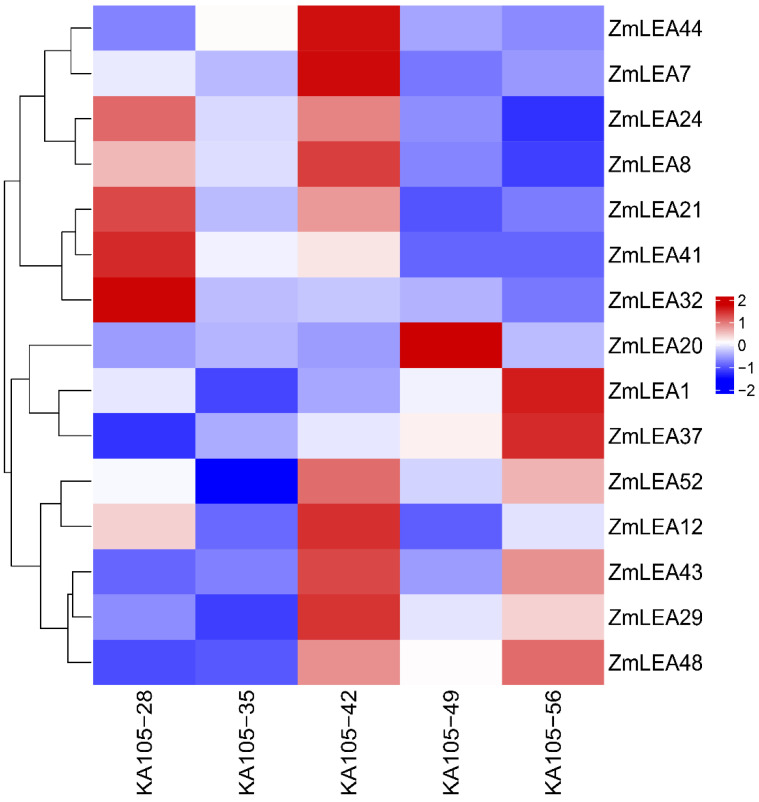
Heatmap of ZmLEA protein expression in KA105.

**Table 1 plants-12-03674-t001:** Characteristics and physicochemical parameters of *ZmLEA* genes in the maize genome.

Gene	Subfamily	Gene ID	Location	Amino Acid	MW (KD)	pI	GRAVY
ZmLEA1	SMP	Zm00001eb004470	chr1:12338113-12339599	291	28,407.97	4.51	−0.209
ZmLEA2	LEA_4	Zm00001eb005250	chr1:14547169-14548604	322	33,788.06	6.03	−0.94
ZmLEA3	LEA_2	Zm00001eb022900	chr1:88936879-88937823	210	22,979.67	10.5	0.16
ZmLEA4	LEA_2	Zm00001eb033830	chr1:187537818-187539807	269	29,952.1	10.24	0.002
ZmLEA5	LEA_2	Zm00001eb044970	chr1:234172861-234174320	274	29,476.16	10.08	−0.104
ZmLEA6	SMP	Zm00001eb047600	chr1:244450281-244451067	238	25,804.11	10.64	−0.634
ZmLEA7	SMP	Zm00001eb057120	chr1:282509952-282511166	266	27,177.35	5.41	−0.367
ZmLEA8	SMP	Zm00001eb057130	chr1:282541392-282542374	205	21,194.35	5.14	−0.53
ZmLEA9	LEA_2	Zm00001eb064120	chr1:303982107-303985376	319	35,274.07	4.92	−0.364
ZmLEA10	LEA_2	Zm00001eb065640	chr2:765492-766740	204	21,790.27	8.9	0.334
ZmLEA11	LEA_2	Zm00001eb066210	chr2:1899747-1900852	251	26,926.11	10.04	−0.064
ZmLEA12	LEA_4	Zm00001eb071160	chr2:11114272-11115597	305	32,277.8	8.39	−0.887
ZmLEA13	LEA_2	Zm00001eb083540	chr2:57422886-57424204	225	24,069.76	7.67	0.313
ZmLEA14	LEA_2	Zm00001eb106290	chr2:210544418-210545275	203	22,724.01	9.1	−0.179
ZmLEA15	LEA_3	Zm00001eb127770	chr3:36494705-36495503	95	9519.67	10.36	−0.001
ZmLEA16	LEA_3	Zm00001eb130570	chr3:57423270-57423663	91	9677.27	9.96	−0.21
ZmLEA17	LEA_3	Zm00001eb130580	chr3:57516852-57517684	101	10,625.25	9.76	−0.22
ZmLEA18	LEA_2	Zm00001eb146910	chr3:181095946-181096962	238	26,325.89	9.4	−0.294
ZmLEA19	LEA_2	Zm00001eb150110	chr3:192024649-192026196	286	30,406.87	10.78	−0.056
ZmLEA20	LEA_4	Zm00001eb155430	chr3:210032549-210033863	231	23,542.86	8.96	−0.763
ZmLEA21	Dehydrin	Zm00001eb155620	chr3:210644563-210645769	236	24,810.07	5.99	−0.893
ZmLEA22	LEA_2	Zm00001eb159430	chr3:223080810-223081905	180	19,631.49	4.83	−0.101
ZmLEA23	LEA_2	Zm00001eb160190	chr3:225307642-225308912	320	32,924.72	11.41	−0.113
ZmLEA24	Dehydrin	Zm00001eb187010	chr4:159677441-159678859	289	31,466.47	5.51	−1.3
ZmLEA25	LEA_2	Zm00001eb198910	chr4:202382893-202385634	213	23,251.76	9.23	0.162
ZmLEA26	LEA_2	Zm00001eb211120	chr5:1888452-1889419	217	23,291.75	7.95	0.226
ZmLEA27	SMP	Zm00001eb213850	chr5:5981874-5982721	219	22,350.03	4.86	−0.255
ZmLEA28	LEA_2	Zm00001eb221020	chr5:26041536-26043597	265	28,327.98	10.26	−0.019
ZmLEA29	LEA_4	Zm00001eb238300	chr5:145131834-145134200	604	61,763.46	7.26	−0.814
ZmLEA30	LEA_2	Zm00001eb243040	chr5:173362167-173363583	207	22,661.28	9.18	0.168
ZmLEA31	LEA_2	Zm00001eb247790	chr5:190986503-190988017	213	23,198.81	10.32	0.074
ZmLEA32	Dehydrin	Zm00001eb250120	chr5:198232923-198234455	290	31,440.76	6.05	−1.25
ZmLEA33	LEA_2	Zm00001eb259940	chr6:5880193-5881185	213	23,414.43	9.14	0.31
ZmLEA34	LEA_2	Zm00001eb259950	chr6:5904344-5905733	214	23,855.72	9.07	0.239
ZmLEA35	Dehydrin	Zm00001eb285360	chr6:148134643-148135795	168	17,075.48	8.78	−1.144
ZmLEA36	LEA_3	Zm00001eb285680	chr6:149412498-149413322	93	10,139.52	6.29	−0.432
ZmLEA37	LEA_5	Zm00001eb286150	chr6:151445811-151446813	113	12,083.08	5.42	−1.206
ZmLEA38	LEA_5	Zm00001eb286170	chr6:151481816-151482230	91	9669.49	6.61	−1.266
ZmLEA39	LEA_2	Zm00001eb304760	chr7:28700129-28701879	321	35,551.82	4.74	−0.433
ZmLEA40	LEA_1	Zm00001eb308610	chr7:78245306-78246134	125	12,633.17	9	−0.477
ZmLEA41	LEA_2	Zm00001eb319570	chr7:149210004-149211074	219	23,006.58	9.32	0.216
ZmLEA42	LEA_2	Zm00001eb323160	chr7:163429460-163430340	203	22,824.46	8.69	−0.046
ZmLEA43	LEA_2	Zm00001eb342130	chr8:61665910-61666724	152	16,088.45	5.64	0.024
ZmLEA44	LEA_2	Zm00001eb344820	chr8:76504841-76505721	166	17,984.54	5.3	−0.093
ZmLEA45	LEA_2	Zm00001eb357160	chr8:142815635-142828474	311	31,816.59	11.2	−0.048
ZmLEA46	LEA_2	Zm00001eb358640	chr8:148997074-148998272	181	19,680.52	4.77	−0.025
ZmLEA47	Dehydrin	Zm00001eb365110	chr8:169507204-169509063	470	50,246.08	9.58	−0.512
ZmLEA48	LEA_4	Zm00001eb395500	chr9:139364531-139366107	354	37,909.35	6.6	−1.036
ZmLEA49	LEA_2	Zm00001eb405330	chr10:1744675-1746249	290	31,093.09	11.55	−0.159
ZmLEA50	LEA_2	Zm00001eb407150	chr10:6543035-6544213	213	23,229.6	8.27	0.135
ZmLEA51	LEA_2	Zm00001eb422110	chr10:117432618-117433773	224	24,042.7	8.83	0.247
ZmLEA52	LEA_2	Zm00001eb434170	chr10:151062372-151063480	221	24,279	8.62	0.113

## Data Availability

The data that supports the findings of this study are available in the Appendix A of this article.

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
