# Peer review of "Identification of the Maize LEA Gene Family and Its Relationship with Kernel Dehydration"

_plants, 2023, doi:10.3390/plants12213674_

Round 1

Reviewer 1 Report

The work "Identification of the maize LEA gene family and analyzing its

relationship with kernel dehydration" requires redrafting of parts:

Introduction: Many sentences are unclear and poorly written. The main errors are in the maize and KMC parts, for example:

- no citation number 2

- mechanization rate - a parameter that is not very clear, is it about the number of farms or the area of ​​crops?

- Currently..- followed by a quote from 12 years ago

- citations 12, 13 refer to Arabidopsis (not maize), it is worth noting

Style: if there is an abbreviation KMC - meaning kernel moisture content, it is worth avoiding the next words "kernel" in this sentence.

- "LEA protein…." and next sentence: "They were named LEA..."

Material and Methods:

No information about the research material: “kernel at different development stages in two imbread lines with varing dehydration rates”.

Results:

It was only on page 3 that the ZmLEA abbreviation was explained - it should have been made at the beginning and unified.

Discussion:

It seems that the authors ran out of ideas for an interesting discussion. Currently, there are so many publications on LEA proteins that the obtained results can be better discussed.

References: not standardized notation.

Improving the English in the manuscript requires making the text more clear and fluent.

Author Response

Response to Reviewer 1 Comments

Thank you for serving as a reviewer to consider our manuscript for publication in Plants. We would like to thank you for the time and effort to review our manuscript. We revised the manuscript thoroughly and made necessary changes asked by the reviewers. Please see below a point-by-point response to reviewers’ comments.

Comments from reviewers:

The work "Identification of the maize LEA gene family and analyzing its relationship with kernel dehydration" requires redrafting of parts:

  1. Introduction: Many sentences are unclear and poorly written. The main errors are in the maize and KMC parts, for example:

- mechanization rate - a parameter that is not very clear, is it about the number of farms or the area of crops?

Response: Sorry for the unclear description, the “mechanization rate” means “Machine sowing rate, ratio of mechanically sown area to total cultivated area”, and the mechanized harvesting rate represented “Ratio of mechanically harvested area to total cultivated area”.

- Currently.- followed by a quote from 12 years ago

Response: Thanks, we have updated all references and corrected language presentation

- citations 12, 13 refer to Arabidopsis (not maize), it is worth nothing

Response: As far as the available research results are concerned, the relationship between LEA proteins and maize kernel dehydration has not reported too less. Therefore, the clear research in model plant Arabidopsis were used as reference to guide that in maize. And from the results in this study, we found there were many similar conclusions in the two species, such as the response of LEA_4 subfamily to dehydration stress.

  1. Style: if there is an abbreviation KMC - meaning kernel moisture content, it is worth avoiding the next words "kernel" in this sentence.

- "LEA protein…." and next sentence: "They were named LEA..."

Response: All terms that can be abbreviated have been abbreviated when they firstly occurred in the body of the text. Also, we check all abbreviations in the whole text.

  1. Material and Methods:

No information about the research material: “kernel at different development stages in two inbred lines with varying dehydration rates”.

Response: Sorry for that. The transcription data of the two maize inbred lines were from previous study (Qu et al. Time-resolved multi-omics analysis of the genetic regulation of maize kernel moisture. The Crop Journal 2023,11(1):247-257 ). The detail Information of the two self-incompatible lines KA105, KB020 mentioned in the text has been added to Materials and Methods in lines 126-144, replacing the simple literature citations in previous version, also the phenotype of these inbred lines in Table S1.

  1. Results:

It was only on page 3 that the ZmLEA abbreviation was explained - it should have been made at the beginning and unified.

Response: According to your advice, we have checked them in the whole manuscript.

  1. Discussion:

It seems that the authors ran out of ideas for an interesting discussion. Currently, there are so many publications on LEA proteins that the obtained results can be better discussed.

Response: Thanks for your advice. The discussion has been rewritten and enriched from the characterization, function of cloned genes or homologous genes, and the application of LEA proteins in the future.

  1. References: not standardized notation.

Response: We have amended the references and checked one by one for the reference.

Reviewer 2 Report

The authors describe an analysis of the LEA gene family in maize and attempt to describe an relationship to kernel dehydration.

I genarally like gene family analyses and characterisation in plant genomes, but this study is weak.

In the results section the authors write that 52 ZmLEA genes were selected for further analysis, where there more LEA genes and why did they not qualify for further analysis?

Figure 2 is too small to read. Figure 2A and 2C need to be independent figures.

How where the subfamilies defined? They are not monophyletic!

Figure 4: what is the left motif? Not all proteins in a family start with the same motif!

Figure 5 is also very small and hard to read.

The expression analysis adds very little to the overall content of the manuscript. Some comparisons with other species would be interesting, is there anything known about homologous genes?

The language is generally ok, but some improvements are possible with restructuring some sentences.

Author Response

Response to Reviewer 2 Comments

Thank you for serving as a reviewer to consider our manuscript for publication in Plants. We would like to thank you for the time and effort to review our manuscript. We revised the manuscript thoroughly and made necessary changes asked by the reviewers. Please see below a point-by-point response to reviewers’ comments.

Comments from reviewers:

  1. In the results section the authors write that 52 ZmLEA genes were selected for further analysis, where there more LEA genes and why did they not qualify for further analysis?

Response: Sorry for the unclear description. Firstly, we obtain 52 LEA genes in maize by compared the maize B73 reference genome with the LEA genes in other species including Arabidopsis, rice and sorghum. Our aim is to identify the relationship between ZmLEAs and kernel dehydration rate, which have not uncovered in maize.  During the motif analysis, the small ZmLEA sub-family only have one or two genes, which have no conservative motif, so we didn’t show them. Also, in the parts of “Differential ZmLEA gene expression in two maize inbred lines with different dehydration rates”, 14 of the 52 ZmLEA genes showed differential expression in the two-target maize inbred lines, so we only exhibited them but not all 52 ZmLEAs.

  1. Figure 2 is too small to read. Figure 2A and 2C need to be independent figures.

Response: Thanks for your advice. We have separated the Figure 2 according to the advice. And check it clear enough to read.

  1. How where the subfamilies defined? They are not monophyletic!

Response: Sorry for the unclear description. For LEA family, the subfamilies have been defined in the model plant-Arabidopsis, based on the sequence homology and conserved motifs in the Pfam database, where the LEA genes were divided into seven subfamilies (Hundertmark, M.; Hincha, D. K. LEA (late embryogenesis abundant) proteins and their encoding genes in Arabidopsis thaliana. BMC Genomics 2008, 9, 118.). The following figure showed the categorization of the LEA family from the article, which is generally accepted. We have improved the description in the Methods and Results part.

PFAM Dure et al Bray Tunnacliffe
and wise
Battaglia et al Bies-Estheve  et al. Hundertmark and Hincha
1989 1993 2007 2008 2008 2008
PF00257(dehydrin) D11 Group2 Group2 Group2 Group2 dehydrin
PF04927(SMP) D34 Group6 Group6 Group5A Group5 SMP
PF03760(LEA_1) D113 Group4 Group4 Group4A Group4 LEA_1
Group4B
PF03168(LEA_2) D95 --- --- Group5C Group7 LEA_2
PF03242(LEA_3) D73 ... Lea5 Group5B Group6 LEA_3
PF02987(LEA_4) D7 Group3 Group3 Group3A Group6 LEA_4
D29 Group5   Group3B    
PF00477(LEA_5) D19 Group1 Group1 Group1 Group1 LEA_5
D132          
PF10714(LEA_6) --- --- --- Group6 Group8 PvLEA18
PF02496(ABA-WDS) --- --- --- Group7 --- ---

4. Figure 4: what is the left motif? Not all proteins in a family start with the same motif!

Response: Sorry for the unclear description. For LEA family, there were different conservative motif between different subfamilies. The left image in figure 4 ( Figure 6 in the new manuscript) shows the motifs of individual genes in each subfamily, with different colors representing different motifs. Also, the motif sequences have been added to the image.

5. Figure 5 is also very small and hard to read.

The expression analysis adds very little to the overall content of the manuscript. Some comparisons with other species would be interesting, is there anything known about homologous genes?

Response: Sorry for that. We have added more information for that in discussion. actually, in the research (Qu,J.Z.; Xu,S.T.; Gou,X.N.; Zhang,H.; Cheng,Q.; Wang,X.Y.; Ma,C.; Xue,J.Q. Time-resolved multiomics analysis of the genetic regulation of maize kernel moisture.The Crop Journal 2023,11(1):247-257.), ZmLEA29 showed significant association with the KDR and KMC. There was little difference in the expression of ZmLEA44 in the transcriptome level during kernel development. And its homologous gene has been reported that it can regulate ABA biosynthesis and metabolism genes through transcriptional level changes (Huang, L.; Zhang, M.; Jia, J.; Zhao, X.; Huang, X.; Ji, E.; Ni, L.; Jiang, M. An Atypical Late Embryogenesis Abundant Protein OsLEA5 Plays a Positive Role in ABA-Induced Antioxidant Defense in Oryza sativa L. Plant Cell Physiol 2018, 59, 916-929.) In addition, the expression of SbLEA3A gene, the homologous of ZmLEA20, was low in developing seeds and increased significantly at maturity. While SbLEA3B-2 was only expressed at maturity (Dalal, M.; Sandeep Kumar, G.; Mayandi, K. Identification and expression analysis of group 3 LEA family genes in sorghum [Sorghum bicolor (L.) Moench]. Acta Physiologiae Plantarum 2012, 35, 979-984.).

6. Comments on the Quality of English Language: The language is generally ok, but some improvements are possible with restructuring some sentences.

Response: Thanks for your advice, we have improved the English language again, hope this type will be better.

Round 2

Reviewer 1 Report

Dear Authors,

Revised version prepared carelessly, ONLY the first few examples:

 Line 7: unnesesery bolded; Maize,

Line 18: Furthermore, theBy analyzing transcriptome data of kernels at different developmental stages in two maize inbred lines from kernels at different developmental stages in two inbred lines with varying dehydration rates, we identifiedshowed that 14 ZmLEA genes expressed that showed differentially expression between the two inbreds.

Line 38:  Therefore, the development of new maize varieties suitable for kernel mechanized kernel harvesting is essential.

Line 35:  mechanizedachine

Line 41: .. kernel mechanized kernel harvest..

Line 47: Late Embryogenesis Abundant proteins 47 (LEA) - the shortcut has already been explained

Line 49: hAlso

Line 49: owever

- quote 7 before 6

- iisolated

- cottondentified in 1981

- They will rapidly synthesize and 57 accumulateMoreover,…

- kernel dehydration rate (KDR) KDR during..

etc..

The work requires serious improvement.

Extensive editing of English language required.

Author Response

Dear editor,
Thank you for serving as editor consider our manuscript for publication in Plants. We would like to thank you and the reviewers for the time and effort to review our manuscript. We revised the manuscript thoroughly and made necessary changes asked by the reviewers. Please see the attachment.

Reviewer 2 Report

Although it is nice to see the changes that are applied to a manuscript, the editing in this case leaves a text that is completely incomprehensible.

incomprehensible text structure

Author Response

Dear editor,
Thank you for serving as editor consider our manuscript for publication in Plants. We would like to thank you for the time and effort to review our manuscript. We revised the manuscript thoroughly and made necessary changes asked by the reviewers. Please see the attachment.

Round 3

Reviewer 1 Report

The improvement has significantly improved the quality of work. Minor editing errors (incorrect spaces, italics). In my opinion, not all sentences sound good linguistically, but I don't want to interfere too much with the original nature of the work.

acceptable level

Author Response

Thank you for your time and effort as a reviewer in reviewing the manuscript for us. Thank you for recognizing the content of our manuscripts.

Reviewer 2 Report

This is a revised version of the manuscript and it has improved, but I am still not happy with the text.

The description of the gene finding has improved, but I still do not agree with the presented result. According to the text, sequences that had been identified were removed from the final set just because they were not annotated in the reference genome, but some of the genes that were retained do not have annotated 5'- or 3'-UTRs, why are they retained and not others?

The splitting into groups is nicely described in the answer to the reviewer, but I did not find an explanation in the text (one sentence with a reference would be enough).

Throughout the manuscript 'ver.' spells 'verb'!

Line 18: 'purifying' not 'purification'

Line 31: 'production' not 'product'

Line 71/325: i don't like the use of 'newest' this might not be the case anymore when the manuscript is published, just use the version number of the reference.

Line 92: remove 'the' in front of 'redundant'.

Line 138: 'freezer' not 'refrigerator'

Line 152/153: chromosomes

Line 202: 'strong' not 'stronger'

Figure 6: The figure legend is still not clearly identifying which motif block in A corresponds to the motif in B. It still says 'left motif', but the left most block in A does not have the same colour in all genes!

 Line 323: the reference has only two authors, so no 'et al.'

English language ok.

Author Response

Thank you for serving as a reviewer to consider our manuscript for publication in Plants. We would like to thank you for the time and effort to review our manuscript. Here, we revised the manuscript thoroughly and made necessary changes. Please see below a point-by-point response to the comments.

1、The description of the gene finding has improved, but I still do not agree with the presented result. According to the text, sequences that had been identified were removed from the final set just because they were not annotated in the reference genome, but some of the genes that were retained do not have annotated 5'- or 3'-UTRs, why are they retained and not others?

Response: Sorry for the unclear description. When identifying ZmLEA, we researched the candidate members in the B73 reference genome by comparing the whole genome using the published LEA protein sequences in sorghum, rice, and Arabidopsis as reference protein sequences. Subsequently, the retrieved candidate genes were checked again by BLAST back to the three species in the SwissProt database (https://expasy.org/resources/uniprotkb-swiss-prot). Finally, the 52 genes, which can be identified in the two methods, were retained as results. The representation of family gene identification has been changed in lines 145~152 of the text.

For the missing 5'- or 3'-UTRs in some genes such as ZmLEA6, ZmLEA16, ZmLEA27, ZmLEA38, we have check again in the two maize database(http://maizegdb.org and http://maizesequence.org), they really don’t have the 5'- or 3'-UTRs. And for genes, not all genes have the 5'- or 3'-UTRs.

2、The splitting into groups is nicely described in the answer to the reviewer, but I did not find an explanation in the text (one sentence with a reference would be enough).

Response: Sorry for the missing description in the text. We have completed the description in lines 153-161 of the article.

3、Figure 6: The figure legend is still not clearly identifying which motif block in A corresponds to the motif in B. It still says 'left motif', but the left most block in A does not have the same color in all genes!

Response: Sorry for the unclear description. We have improved the Figure 6 in the manuscript and the related file “Figures and Legends”. (Figure 6. Conserved motif map of the ZmLEA gene. Conserved motif in each ZmLEAs and representative sequence of the consensus motif in corresponding subfamily.)

4、Other issues about the expression (Throughout the manuscript 'ver.' spells 'verb'!; Line 18: 'purifying' not 'purification'; Line 31: 'production' not 'product'; Line 92: remove 'the' in front of 'redundant'; Line 138: 'freezer' not 'refrigerator'; Line 152/153: chromosomes; Line 202: 'strong' not 'stronger'; Line 323: the reference has only two authors, so no 'et al.'.)

Response: Deeply sorry for any inappropriate words. We have corrected the inaccuracies you pointed out in the text and have thoroughly checked the article.

Round 4

Reviewer 2 Report

This version has improved again, but Figure 6 is still not OK.
It is still unclear which colored block corresponds to the presented motif.

I would suggest to remove the motifs from the figure. They do not add any useful information to the manuscript!

Line 87 contains still a verb5 instead of version 5!

Author Response

Dear reviewer.
Thank you for your proposed revisions, which we have refined. Please review them in the attached document.

Round 5

Reviewer 2 Report

Figure 6 could be improved by using all groups.

Otherwise the manuscript is acceptable now.

Author Response

Question: Figure 6 could be improved by using all groups.
Re:
Dear reviewer,
Due to the small number of other subfamily members, no representative motifs were found during the analysis. Thank you for your valuable time and effort in reviewing the article.